# Enhancing Detection Quality Rate with a Combined HOG and CNN for Real-Time Multiple Object Tracking across Non-Overlapping Multiple Cameras

**DOI:** 10.3390/s22062123

**Published:** 2022-03-09

**Authors:** Lesole Kalake, Yanqiu Dong, Wanggen Wan, Li Hou

**Affiliations:** 1School of Communications and Information Engineering, Institute of Smart City, Shanghai University, Shanghai 200444, China; yanqiu_dong@shu.edu.cn (Y.D.); wanwg@staff.shu.edu.cn (W.W.); 2School of Information Engineering, Huangshan University, Huangshan 245041, China; houli@shu.edu.cn

**Keywords:** convolutional neural network, histogram oriented graphic, multi-camera multi-object tracking, detection quality

## Abstract

Multi-object tracking in video surveillance is subjected to illumination variation, blurring, motion, and similarity variations during the identification process in real-world practice. The previously proposed applications have difficulties in learning the appearances and differentiating the objects from sundry detections. They mostly rely heavily on local features and tend to lose vital global structured features such as contour features. This contributes to their inability to accurately detect, classify or distinguish the fooling images. In this paper, we propose a paradigm aimed at eliminating these tracking difficulties by enhancing the detection quality rate through the combination of a convolutional neural network (CNN) and a histogram of oriented gradient (HOG) descriptor. We trained the algorithm with an input of 120 × 32 images size and cleaned and converted them into binary for reducing the numbers of false positives. In testing, we eliminated the background on frames size and applied morphological operations and Laplacian of Gaussian model (LOG) mixture after blobs. The images further underwent feature extraction and computation with the HOG descriptor to simplify the structural information of the objects in the captured video images. We stored the appearance features in an array and passed them into the network (CNN) for further processing. We have applied and evaluated our algorithm for real-time multiple object tracking on various city streets using EPFL multi-camera pedestrian datasets. The experimental results illustrate that our proposed technique improves the detection rate and data associations. Our algorithm outperformed the online state-of-the-art approach by recording the highest in precisions and specificity rates.

## 1. Introduction

The visualization and tracking of multiple objects in surveillance applications are enormously dominating topics in computer vision’s security field. In recent years, there has been a drastic change in point of focus for enhancing the handling of security issues on these applications [1]. Many researchers are attracted, and several techniques and algorithms emerged are applied continuously on various smart city projects to ensure residence safety. However, most rely on the traditional convolutional neural network (CNN) to improve the detection quality rate and object classification [2]. The CNN provides an effective and quick solution to extract high-level contour features and record a significant state-of-the-art performance on real-time multiple-object-tacking (MOT) [3]. It is considered to be more effective compared to HOG descriptor algorithms which mainly focus on global features process handling [4].

Despite the state-of-the-art achievement, the traditional CNN proposed algorithms tend to ignore the global features [5]. Their detectors are mainly based on the local features extraction for the application to understand the image information [6]. Therefore, they continue to suffer from identifying the shape and boundary characteristics from the captured images [7]. Thus, this contributes to their incapability for handling the detection accuracy on light, appearance distortion, deformation, and motion-blurred images. Furthermore, it results in poor detection quality and high false positives, hence, its failure in representing human-like application systems [8]. Other studies tried to eliminate this grey area by exploiting the HOG descriptor technique and recorded satisfactory results but suffered from the speed and classification of huge samples during the training phase [9].

Therefore, to ensure both contour and global features are effectively incorporated into the neural network to represent a human-like system. In this paper, we propose to build a new model by combining the HOG descriptors and a traditional CNN to form an HCNN algorithm for tracking multi-object across non-overlapping cameras. We further propose to improve the detection quality rate by removing the background information and ensuring that the appearance and motion variations are well maintained throughout the tracking process. This paper is arranged into five sections: Section 1 introduces the background, Section 2 details the related work, Section 3 describes details of our approach, Section 4 presents experimental results, Section 5 discusses an interpretation of the results and comparison with state-of-the-art algorithms, and finally Section 6 concludes the paper.

## 2. Related Works

The techniques that implement multiple view angles provide additional information that enables the computer vision applications to acquire more knowledge and understanding of the object’s characteristics. This has proven its effectiveness in enriching the target-related shape, features, and location in sequential video frames [3]. It further resulted in the emergence of various multiple view object tracking approaches to solve the persisting challenges such as partial inclusion, shape deformation, illumination variations, and background cluttering. The approaches are online or offline depending on the criteria, such as handcrafted features or deep features handlings. The handcrafted feature-based trackers are manually defined, whereas the deep features trackers use neural networks [10]. However, both categories tend to ignore the preprocessing of input images to reduce interferences. Therefore, integration has emerged to achieve fast and accurate human-like detection application systems [11]. Thus, in this section, we summarize these previously proposed state-of-the-art tracking methods by classifying them into two themes: (i) histogram of oriented gradient (HOG) and (ii) convolutional neural network (CNN) learning-based methods.

The histogram of oriented gradient (HOG) descriptor is one of the most popular approaches in computer vision used to extract significant features from images. It discards the futile information by relying heavily on the extracted features to compute accurate objects detections and classifications [12].

Zhang et al. [11] were inspired by these capabilities and proposed a combined local and global feature handling algorithm to simulate a human-like application. They trained both features (local and global) with traditional CNN and set the number of hidden layer nodes to 3000 to distinguish the fool images. However, the technique is most efficient in offline mode and recorded few false alarms compared to CNN solely based paradigms. It further illustrated the incapability of learning features recursively and resulted in slow detection performance, decreased accuracy, and posed challenges to implement online. To eliminate these challenges, Zhang et al. [13] introduced the model detection and classification of moving objects in video and used HOG to remove the noisy background. This strengthened the approach in detecting the moving objects accurately in food and agricultural traceability analysis. However, it failed to obtain adequate features from the selection and resulted in a poor detection rate and data association. Najva et al. [14] proposed improving the detection rate by combining tensor features with scale invariant feature transform (SIFT) features. The technique merged the handcrafted features with a deep convolutional neural network (DCNN) and served as the concrete foundation to expand in the computer vision field. Then Lipetski et al. [5] took advantage of the laid foundation and combined the HOG descriptor with CNN to form the HCNN model for improving the pedestrian detection quality rate. They extracted HOG features and fed them into the CNN as input to increase classification and detection rates. This reduced the processing time of the overall detector and proved that the concept enhances the capabilities of the overlapping window to handle real-time object tracking processes. The development gained the attention of Rui et al. [15], who proposed an algorithm that takes various features maps from the first CNN layer as input to HOG and extracts the HOG features. However, the performance results illustrated that a single feature map was not comprehensive enough to reflect all the necessary information on the original image. Thus, the technique performed worse than the original HOG paradigm but proved that pedestrian detection with HOG-based multi-convolutional features could obtain a high detection accuracy and stabilized network performance. Then Sujanaa et al. [16] proposed to eliminate pedestrian detection and classification issues by introducing the combined pyramid histogram of oriented gradient (PHOG) and CNN algorithm for real-time object tracking. They used the PHOG descriptor to create pyramid histograms over the entire image and attach them into a single vector, whereas the CNN is used as the classifier for the PHOG features extracted from the window’s raw image data. The first layer of the CNN moved adequately over the input image window thus that the second layer could transfer functions to the input image window. Lastly, the hidden layer unit is used to connect to each input through a separate weight. This reduced computational cost, adaptable parameters during training, and proved the technique compatible for real-time object tracking. However, it suffered from low performance with a high misdetection rate under heavy light variations.

Qi et al. [17] proposed an internet of things (IoT) based on a key frame extraction algorithm to enhance detection quality rate in videos. They modeled and trained the CNN to generate a predicted score to indicate the quality of faces in the frame. The selected key frames fed into the neural network to enhance face detection accuracy. This enhanced the extraction of feature vectors and increased face recognitions and detections on poor-quality captured images. Angeline et al. [1] capitalized on the progress and proposed to enhance efficiency on face recognition applications in real-time object tracking. They used HOG descriptor detections to enhance accuracy and train CNN with a linear support vector machine (SVM) to handle blurred motions, occlusions, and pose variation. However, the algorithm used a small dataset and struggled with misfeeding. Thus, Yudin et al. [18] used video streams of specified IP cameras to access more data through the server module. They augmented the IoT application with the HOG descriptor and masked R-CNN architecture for accurate detection of a human head on low-quality and light variations images. This enabled the application to carry out client requests from various computers connected to the network. However, the updating of people counting results performed once per minute hindered overall speed performance. This contributed to the misdetection rate where objects’ motion changes.

Madan et al. [6] proposed a hybrid model based on a combination of HOG-speeded-up robust features (SURF) features and CNN. They used extracted HOG features as an input into the network (CNN) and reduced the dimensions. The application embedding from the first layer and second layer of the CNN passes through the fully connected layer. Therefore, this reduced the model parameter’s computational cost by filtering out the fool images at an early stage. It further improved the detection and classification accuracy rate. Bao et al. [7] showed appreciation of these developments when proposing the merging of both HOG feature space and traditional CNN to ease the plant species identification and classification from a leaf pattern in botany. They extracted HOG features through 8 × 8 dimension cells and 2 × 2 cells per block for the input image. These attributes are passed into the network for further processing and classification. However, the algorithm is an offline mode and recorded a noticeable improvement in the overall performance.

## 3. Proposed HCNN for Real-Time MOT

The main task is to track and re-identify the target across these multiple cameras [19,20,21]. We, therefore, designed our algorithm to detect, track and re-identify the object of interest across several non-overlapping cameras using the multi-object tracking process. We implemented the proposed algorithm using the dataset that contains different poses of persons [22] and different illumination conditions. The algorithm is divided into two modules, namely, detection and tracking. The detection module buttressed [23,24] by the inclusion of HOG descriptors which have been proven to cater to both texture and contour features [8,21,22]. We train the model on the EPFL dataset with multiple pedestrians’ videos using the HOG detector. However, the HOG descriptor is slowing down overall algorithm performance. Therefore, we combined the HOG detector module with CNN to create an HCNN to enhance classification and identify the association in tracking multiple people. According to our best knowledge, there is no similar proposed algorithm for real-time object tracking across multiple non-overlapping cameras.

The algorithm’s process of determining an object’s background is split into several separated steps to eliminate backgrounds that might otherwise be classified [25,26,27]. This is embraced by subtracting the objects’ background and computing a foreground mask on colored video frames [28] and gray images captured from multiple surveillance cameras. The proposed algorithm takes an input of the 120 × 32 images, cleans and converts them into binary format, and then smoothens the pixels on binary images by applying morphological operations that are followed by the implementation of a Laplacian of Gaussian model (LOG) mixture after blobs. The images undergo further feature extraction and computation with the HOG descriptor. We stored these features into a 2-dimensional array and passed them to the fully connected multi-layer neural network for further classifications and matching computation, as shown in Figure 1. The CNN flattens the given 2D array into a single feature vector that is used to determine the object of interest’s class. Then an output from the HOG descriptor compared them with the object of interest on the input frame based on the connected components, image region properties, and window binary mask. The sliding window tactics were applied on input frames to reduce the data size, processing time, and to improve the object locating during tracking in one step. The normalized cross-function is used to obtain the object centroids on these images. Finally, we considered the use of the Kalman filter to track the object of interest, based on the computed centroids.

### 3.1. Background Segmenting Modeling

Background motion has always been a throwback for many conventional methods to achieve the desired accuracy [23,25]. However, we applied the background subtraction model to ensure that our algorithm overcomes these challenges. In this modeling, we set the threshold pixel value to 0.5 to ensure the detection of every blob for all objects shapes. We further applied the splitting of the image into foreground and background for our algorithm to efficiently classify the pixels [29]. However, the recent history of each pixel value is observed with a mixture of Gaussian distributions, and the new pixel values are considered as the major components to update the model. These new pixel values at a given time ($pV_{t}$) are further checked against generated Gaussian distribution until matches are obtained [30]. The pixels with similar velocity at given x and y directions are considered as a point of interest of the same object representing its velocity. These matches are then defined with a standard deviation (σ) of the distribution. This improved the foreground masks, connectivity between neighboring pixels, speed mapping of the moving object, and the capability to distinguish the non-stationary detections from the foreground blobs.

However, when there are no matches found in the T generated distribution, the probability of the distribution of the previous action is replaced with the current mean (μ) value, highest variance ($\sigma^{2}$) and the lowest weight (w) of the object. Thus, we observe the probability of the pixel values as follows.
(1)$$P\left(X_{t}\right)=\overset{T}{\underset{i=1}{\sum }}W_{i,t}\times \Psi (X_{t},\;\mu_{i,t},\overset{T}{\underset{i,t}{\sum }})\;$$
where $\left\{X_{1},X_{2}\ldots X_{t}\right\}$ represent recent pixels history and 1≤i≤t; *T* denotes the number of the distributions, whereas $W_{i,t}$ represents an estimated weight of the *i*th The Gaussian mixture at given time t, $\mu_{i,t}$ and $\overset{T}{\underset{i,t}{\sum }}$ respectively denotes the mean and covariance matrix. Then Ψ denotes the Gaussian probability density function and is computed as follows.
(2)$$\Psi \left(X_{t},\mu ,\Sigma \right)=1/({\left(2\Pi \right)}^{n/2}|\Sigma {|}^{1/2})e^{1/2{\left(X_{t}-\mu_{t}\right)}^{T}\Sigma^{-1}\left(X_{t}-\mu_{t}\right)}\;$$

Then the weight of the T distribution at a given time is updated as follows.
(3)$$W_{T,t}=\left(1-\alpha \right)\times W_{T,\left(t-1\right)}+\alpha \times \left(\Xi_{T,t}\right)\;$$
where α denote the learning rate, *T* is equivalent to the available memory and computation power usage,${\;\Xi }_{T,t}\;\epsilon \left(1,0\right)$ where one denotes model matching is true, zero represents that model as unmatched. The advantage of this background technique we applied is that our background model is updated without destroying the existing model. This is achieved by ensuring that after the weights normalizations, the mean and the variance corresponds with the conditions of the distribution and are updated only when conditions change by using the following equations, respectively.
(4)$$\mu_{t}=\left(1-p\right)\mu_{\left(t-1\right)}+pX_{t}\;$$
and
(5)$$\sigma^{2}{}_{t}=\left(1-p\right)\sigma^{2}{}_{\left(t-1\right)}+p\left(X_{t}-\mu_{t}\right)\;$$

We further ensured that the learning factor *p* adapts to the current distributions by computing it as:(6)$$p=\alpha \times \Psi (X_{t}|\mu_{t},\sigma_{t})\;$$

### 3.2. Foreground Blobs Windowing Modeling

In this modeling, we applied a sliding window approach on both images and foreground frames. This helped our algorithm to avoid detection of non-moving background and shadows of the objects in motion. Therefore, the binary foreground image is used to fulfill the desired window output that is extracted and expressed with the following equation.
(7)$$\zeta_{xy}^{w}=\{\delta_{xy}^{w}\;\;\Sigma_{x,y=1}^{w_{x}w_{y}}\delta_{xy}^{wb}\geq \kappa_{p}\times 50\%;\;\;w_{x}\epsilon \mathrm{hieght},\;w_{y}\epsilon \mathrm{width}$$
where $\zeta_{xy}^{w}$ denotes desired output window for the given window input image $\delta_{xy}^{w}$ which is extracted through a sliding window on the binary foreground image $\delta_{xy}^{wb}$ with a sum of the total number of pixels $\;\Sigma_{x,y=1}^{w_{x}w_{y}}\delta_{xy}^{wb}$ on binary window $\kappa_{p}$. In the next section, we discuss the HoG descriptor implemented in this paper in detail.

### 3.3. HOG Descriptor’s Features Extraction

Hog is a feature extraction technique that extracts features from every position of the image by constructing logic histograms of the object from the images [7]. In this paper, the images are first passed through the HOG descriptor for data size reduction and searching for an object to detect. Thereafter, the histograms are created and computed over the whole images that are retrieved from several video frames. These histograms are then appended into a single feature vector using the exponential equation $2^{\ell }$, representing the grid level (ℓ) for all cells along the dimensions. However, the correspondence on the whole input images between the vectors and histograms bins is ensured by limiting the level (ℓ) to ≤3 and computed using the following equation.
(8)$$\nu =K\sum_{i=1}^{\ell }4^{\ell };\;\;i\leq 3\;$$
where ν, denotes vector dimensions, K denotes bins, ℓ defines grid level. This equation ensures that all images that are extremely large and rich in texture are weighted the same as low texture images within the set parameters. It is also used to guard and control our algorithm against overfitting.

In our detection module, a two-dimensional (2D) array of the detected object is constructed. It is passed to the CNN, wherein the process of targeted object recognition is flattened into a single vector using two fully connected layers. The CNN is also used to classify that the person detected by the HOG descriptor is either associated with the assigned ID (e.g., ID1 or other IDs) [31].

### 3.4. Structure of the Convolutional Neural Network

The structure of the CNN incorporated into our algorithm is shown in Figure 2. We considered extracting the appropriate features first from the window’s raw data. There are four convolutional layers with three max-pooling layers, two fully convolutional layers, and a softmax activation function. The first layer is used to map various small features that are cited as local receptive fields (LRF) that move satisfactorily over the input image window on the grid. The second layer contains one or many fully connected output neurons that are applied to transfer functions to the inputs during the training phase. Therefore, the hidden layer of the multiple layer perception is used to connect each input with a separate weight.

The LRF was applied to all image portions using the same weights, and this contributed to the reduction of adaptable parameters. However, when the network has biased weights, the output weights becomes the element of the transferred functions, which are applied to the first and second layer, respectively. The object is then recognized from the foreground frame’s sliding window, and its parameters such as x and y coordinates for the starting position, height, width, and centroids are calculated. This avoided network overfitting and provided the current location of the object being detected [24,25]. Finally, the Kalman filter was applied to track the object of interest based on the computed centroids and assigned unique identities throughout the frames.

### 3.5. Designing Kalman Filter for Our HCNN Algorithm

In most cases, computer vision algorithms’ frequent task is based on object detection and localization [26,32]. Therefore, in this paper, we considered the design and the incorporation of a simple and robust procedure to engage complex scenes with minimum resources [27]. We integrated the computed object centroids into the Kalman filter’s object motion and measurement noise [29]. This strengthened the processing of noises and the estimation of the object’s next position in the next frame at a given speed and time [12]. However, it also made our algorithm entitled to efficiently re-detect the moving object during occlusions, scaling, illuminations, appearance changes, and rapid motion on both training and validation phases [33]. Therefore, to solve these challenges, we enabled the Kalman filer to model and associate the target ID that is assigned based on the computed centroids. This improved the observations, predictions, measurements, corrections, and updating of the object’s whereabouts and directions.

Thus, observations are effectively used to locate the object and provide a direction at a given velocity and measurement using the following equation.
(9)$$Z=X+{ℰ}_{r\;}\;;\;$$
where Z denotes measurements, X represents the location of the object being tracked, and ${ℰ}_{r\;}$ is distributed normally (${ℰ}_{r\;}$~*N* (0,${\;\sigma }^{2})$) and denotes noisy measurements due to uncertainty of the current object location. Although this guarantees that our algorithm can handle the noises, we prognosticate that our detector might be imperfect due to the combination of ${ℰ}_{r\;}$ and velocity (v) variations that will affect the tracker to locate and track the object of interest effectively. Thus, to handle these uncertainties, we estimated the trajectories of the moving object from the initial state to the final state of direction by incorporating the ${ℰ}_{r\;}$ into the converted matrix formulae of motion measurement as follows.
(10)$${\bar{X}}_{t}=\left[\begin{array}{c} X_{t}\\ V_{t} \end{array}\right];$$
denoting location *X*, and speed *V* of an object at a particular time
(11)$$\;\bar{Z_{t}}=\left[Z_{t}\right];$$
denoting the distance measurement of an object at a particular time

Thus, the Equations (10) and (11) are combined and expanded to express the location of an object being tracked as follows:(12)$$Z_{t}=X_{t}+{ℰ}_{r}\;$$
which is further converted into a matrix equation and used to handle both noisy measurements and speed variation.
(13)$${\bar{Z}}_{t+1}=\left[\begin{array}{cc} 1 & 0 \end{array}\right]{\bar{X}}_{t}+\bar{{ℰ}_{r}};$$
$\left[\begin{array}{cc} 1 & 0 \end{array}\right]$ denote *H* state control matrix at time *t* + 1.

In short, Equation (13) is expressed as ${\bar{Z}}_{t+1}=H{\bar{X}}_{t}+\bar{{ℰ}_{r}}\;.$

However, the Equation (13) estimations do not adapt to the speed changes. Therefore, to incorporate speed variations and locate the position of the object correctly in the next frame, we calculated the algorithm evaluation through time (t) at acceleration (a) and changes in time (Δt) using the equation below.
(14)$$X_{t+1}=X_{t}+V_{t}\times \Delta t+\frac{1}{2}at^{2}\;$$
where $X_{t+1}$ denotes our prediction corrections, $X_{t}$ denotes the location of the object at a given time (t),${\;V}_{t}$ denote the speed of the object at a given time (t), and $\Delta t+\frac{1}{2}at^{2}$ represent speed integration at a given time (t). However, the speed is not constant for the object in motion. Hence, we accommodated its changes through different frames scenes by adapting velocity variations using the equation below.
(15)$$V_{t+1}=V_{t}+a\Delta t$$

We further expanded Equation (14) for time evolution handling and to ensure that the motion and object feature representation on both foreground frames and binary images are correctly captured and predicted. Hence, the newly desired formulae are expressed as follows:(16)$${\bar{X}}_{t+1}=\left[\begin{array}{cc} 1 & \Delta t\\ 0 & 1 \end{array}\right]+{\bar{X}}_{t}\left\{\mathrm{Previous}\;\mathrm{state}\right\}+\left[\begin{array}{c} \frac{1}{2}\Delta t^{2}\\ \Delta_{t} \end{array}\right]a;\;$$
where $\;\left[\begin{array}{cc} 1 & \Delta t\\ 0 & 1 \end{array}\right]$ denotes state transition matrix function (*F*), a denote object’s acceleration and is distributed normally with mean 0 and variance of the noise measurements, $a∼N\left(0,\sigma_{r}^{2}\right)$. Therefore, Equation (16) is further expressed in short, as ${\bar{X}}_{t+1}=F{\bar{X}}_{t}+GV_{t}$ where *G* represents a vector $\left[\begin{array}{c} \frac{1}{2}\Delta t^{2}\\ \Delta_{t} \end{array}\right]$, which is the object’s uncertainty in time changes. Finally, we used these equations into the Kalman filter to predict and correct the object velocity based on the pixels found in the *x* and *y* directions. We predicted the steps and propagated the state as follows:(17)$${\bar{X}}_{t}\Rightarrow {\bar{X}}_{t+1},$$
(17a)$$if({\bar{X}}_{t}∼N({\hat{\bar{X}}}_{t},{\overset{´}{P}}_{t}))$$
where $X_{t}$ is a random variable of a normal distribution with a mean ${\hat{\bar{X}}}_{t}$ and covariance ${\overset{´}{P}}_{t}$.
(17b)$$then\;{\hat{\bar{X}}}_{t+1}=F\cdot \;{\hat{\bar{X}}}_{t}$$
where F represents the previous state with a certain speed at a particular time. Therefore, we expanded the covariance equation to estimate and update time as follows.
(17c)$$P_{t+1}=FP_{t}F^{T}+G\sigma_{a}^{2}G^{T}$$
where $P_{t+1}$ defines the estimated error covariance matrix in the next frame. Thus, knowledge of the measurement ($Z_{t})$ steps are now incorporated into the moving object’s estimate state vector (${\bar{X}}_{t}$) and the (a) Measuring residual error, (b) Residual covariance, and (c) Kalman gain are computed as respectively as follows.
(18a)$$\bar{Y}={\bar{Z}}_{t}-H\cdot {\hat{\bar{X}}}_{t}\;;\;{\hat{\bar{X}}}_{t}=\mu$$
(18b)$$S_{t}=HP_{t}H^{T}+R;\;\mathrm{where}\;R\;\mathrm{denote}\;\sigma_{r}^{2}$$
(18c)$$K=P_{t}H^{T}S_{k}^{-1}$$

Therefore, after this measurement steps incorporation, we can finally update the variable position estimates in the next frame by updating the mean and covariance based on the Kalman gain using the equations below.
(18d)$${\hat{\bar{X}}}_{|z}={\hat{\bar{X}}}_{t}+K\cdot \bar{Y};\;\mathrm{where}\;{\hat{\bar{X}}}_{t}\;\mathrm{denote}\;\mathrm{the}\;\mathrm{previous}\;\mathrm{mean}$$
(18e)$$P_{|z}=\left(I-K\cdot H\right)P_{t\;};$$
and *I* is a 4 × 4 identity matrix: $\left[\begin{array}{cccc} 1 & 0 & 0 & 0\\ 0 & 1 & 0 & 0\\ 0 & 0 & 1 & 0\\ 0 & 0 & 0 & 1 \end{array}\right]$.

## 4. Experiments

### Experimental Setup

We performed experiments on the EPFL datasets based on campus passengers and subway scenes that contain lots of poses and illumination variations. The algorithm is implemented on Dell, G15 Corei7 11800H Processor, NVidia GeForce RT 350Ti GPU, 4 GB GDDR6, 16 GB RAM with Python 3 (Dell, Pretoria, South Africa).

**Datasets and Evaluation Metrics:** The EPFL dataset is used and contains campus and passageway scenes that are both outdoor sequences. The campus scene consists of 6 videos, while the passageway has 4 videos. The videos are split into training and validation sets, where we selected 4 campus scenes videos, 3 passageway videos and split them into frames, and retrieved 40,000 images for training. The remaining videos are used for validation in the testing phase.

The algorithm training is conducted with 30,000 multi-view angle positive images and 10,000 negative images of size 120 × 32. These images are subsets of the frames of the video. We show their instances, labels associations, and correlations in Figure 3. The algorithm is trained with the use of the HOG descriptor, which resized images and activated the object detection module. The HOG descriptor is integrated with the structured CNN illustrated in Figure 2 that is applied as an additional processing mechanism and also a classification mechanism. The training of this proposed system was conducted with 3000 iterations at a learning rate of 0.001.

We evaluated our algorithm’s performance with CLEAR MOT metrics that include the precisions(P), recall(R), identity F1 score(IDF1), mean average precisions(mAP), multiple object tracking accuracy(MOTA), multiple object tracking precisions(MOTP), mostly tracked(ML), mostly lost(ML) and ID switches(IDs). The P is the ratio of the correct positive predictions out of all the positive predictions made, whereas R is the ratio of the number of correct positive predictions made out of all positive predictions that could have been made. The mAP was used to evaluate our detection model by comparing the ground truth-bounding box with the detected box. However, the MT and ML account for the ground-truth trajectories that are the ratio of 80% and 20% correctly identified detections over the mAP returned scores respectively [28]. These metrics are defined as follows:(19)$$\mathrm{Precision}=\frac{\mathrm{TruePositives}}{\left(\mathrm{TruePositives}+\mathrm{FalsePositives}\right)}$$
(20)$$\mathrm{Recall}=\frac{\mathrm{TruePositives}}{\left(\mathrm{TruePositives}+\mathrm{FalseNegatives}\right)}$$
(21)$$\mathrm{IDF}1\;\mathrm{scores}=2\left[\frac{P\times R}{\left(P+R\right)}\right]$$
where *P* and *R* denote precision and recall respectively.
(22)$$mAP=\frac{1}{2}\sum_{k=1}^{k=n}AP_{k}\;;\;AP=\sum_{k=0}^{k=n-1}\left[R_{k}-R_{k+1}\right]\times P_{k}$$
where $R_{n}\;$= 0, $P_{n}\;$= 1 and n denotes the number of thresholds. The k represents the number of classes.
(23)$$\mathrm{MOTA}=1-\left[\frac{\sum_{t}^{N}\left(fn_{t}+fp_{t}+IDs_{t}\right)}{\sum_{t}^{N}G_{t}}\right]$$
where $fn_{t}$, $fp_{t}$ and $IDs_{t}$ denote the number of false-negative or missed detections, the false positive, and the miss-match errors in frame t. The $G_{t}$ represent the ground truth.
(24)$$\mathrm{MOTP}=1-\left[\frac{\sum_{i,t}^{N}d_{t}^{i}}{\sum_{t}^{N}c_{t}}\right]$$
where $d_{t}^{i}$ denotes the distance between the localization of objects in the *i*th ground truth and the detection output in frame t. The $c_{t}$ is the total matches made between ground truth and the detection output in frame t.

**Parameter Settings.** Our Algorithm reacted to a new entry object by initiating a Kalman filter for object tracking [29]. The tracker continues to track and check if the new object falls within the acceptance region of the trajectories by using the Kalman filter predicting equations [12]. The error between the actual observation and the predicted observation is normalized by the computation of a covariance matrix from the Kalman filter update equations [32]. Thus, the determination of whether the new object observation is associated with an existing track is performed by the threshold value test on the residual error (covariance matrix values) [12]. This defines the acceptance relations for each object being tracked and updates the state where the threshold test satisfies. All trajectories that are shorter than 80 milliseconds are deleted. However, when an object observation does not fall within any acceptable trajectory region, the tracker establishes a new track. This endorsed the auto-labeling correlations showed in Figure 3a,b and Figure 4a,b. Therefore, the instances are only associated with a single label, and this has increased the label correlations, precisions, and recall in our experimented dataset [30,31]. It also led to the highest MOTA and MOTP, as shown in Table 1, Table 2 and Table 3. The metrics results and analysis are discussed in the next section, Results Analysis.

## 5. Results Analysis

In this section, we analyze our HCNN algorithm’s results obtained from the experimented dataset. We trained and evaluated our detector to classify with a coupled HOG descriptor and CNN using the EPFL dataset with the selected scenes (campus and passage) for real-time multi-object tracking. The objects are observed and tracked by use of Kalman Filter, as shown in Figure 1. Figure 5, Figure 6, Figure 7 and Figure 8 illustrate the overall performance and effectiveness of our algorithm’s detector and classify for both training and validation phases. 

The algorithm has proven to be effective with high performance in precision and recall, accompanied by the high confidence values on the campus scene dataset. It achieved a greater balance between precision and recall, with a mean average precision of 95.1% at a 0.5 threshold for all classes. This demonstrated in Figure 9 that the algorithm could be trusted for accurately detecting and correctly classifying the objects of interest. However, through this process, the algorithm at the beginning of training and the testing phases had challenges of the unrepresentative data but gradually converged well with more training epochs. This is shown in Figure 6 and Figure 8, with the ups and downs of the jumping of the stats values in either training or validation phase graphs. Thus, it led to the high numbers of false-positive classification and miss matching as clearly advocated in Figure 5 and Figure 7, and Table 1, Table 2 and Table 3. It is emphasized in Figure 6 and Figure 8, where the algorithm training losses and gains on 200 and 100 epochs are projecting the performance well on both the campus and passageway sequences scenes, respectively.

However, the algorithm demonstrated better performance on passageway scenes, which had more difficult challenges such as illumination variations, and different poses compared to the outdoor environment (campus scenes). This is well illustrated in Figure 6 and Figure 8 performance comparisons, where our algorithm recorded the highest performance in precision, recall, and IDF1 scores on the passageway scenes dataset than on the campus scenes dataset. It recorded an absolute 100% for all those metrics with satisfactory confidence values. It is illustrated in Figure A1a,b that our algorithm has mostly identified all the objects of interest under various heavy conditions [32]. This proves that the algorithm is robust against various heavy illuminations and different poses or skewed view angles. However, Figure 9 shows that though the algorithm performed better, it had similar challenges of the unrepresentative data, mostly in the middle of training and testing phases. However, it quickly converged better compared to campus scenes. This proved that our algorithm in the training phase had been fitted with enough data, although at the beginning of our training on the campus scenes, it could be seen struggling or not receiving enough data. The up and downs jumping [33] could be due to data fit because we can see that when we trained the algorithm with more epochs, we obtained better and more stable results for both passageway and campus scenes datasets.

To demonstrate our algorithm’s classification accuracy (CA) and specificity, we compared our precision results with state-of-the-art paradigms. The results are summarized in Table 1. Our approach achieved better results compared to the online approach and short just 5.74% to the current state-of-the-art paradigm.

Thus, for real-time tracking, we evaluated our algorithm with several video frames taken from two different sequences of the EPFL dataset, as shown in Table 2 and Table 3. The CLEAR MOT is used for evaluations, where ↑ denotes high performance and ↓ represents lower performance. In both sequences, our approach recorded an average overall performance above 80% with very few fragmentations and ID switches in all metrics. Further training and testing were conducted on our algorithm without Kalman filter using the 8000 frames from the real-time overlapping multiple cameras dataset (EPFL-RCL multi-cameras). In the comparison exercise, we found that the model’s MOTA, MOTP, precision, and recall performance were very low compared to the one with the Kalman filter in Table 2, Table 3 and Table 4. It had a low detection ratio and a high ID switches ratio that adversely affected the overall tracking results. This is displayed in Table 5 and illustrated well in Figure 9e,f, where the Kalman Filter and segmentation technique are removed from our proposed HCNN algorithm. However, the proposed HCNN with Kalman filter performed very closely to the Yolo5Deep model in Table 4. This proves that the proposed model provides a better data affinity of a close equivalent to the Yolo5Deep model in real-time multiple object tracking.

### Benchmark Evaluation Results

Results on EPFL multi-camera pedestrian datasets: In Table 6, we summarized the results of the EPFL multi-camera pedestrians tracking testing set. We compared our algorithm to several state-of-the-art methods. However, some of these approaches could only be analyzed offline.

For the offline mode, our approach performs poorly. Interestingly, we found that in real-time tracking settings, our approach recorded results that were close to the best state-of-the-art approach. However, in ablation studies, as shown in Figure 9e,f, our approach suffered from overlapping detection boxes and resulted in high misdetection and object re-identification.

## 6. Conclusions

Our study presents an efficient algorithm for multi-view pedestrian detection, identification, and tracking based on combined HOG descriptors and CNN. The background subtraction technique was used to eliminate noise from video frames taken from the EPFL dataset. Extensive experiments were conducted on selected sequences (campus and passageway) of the outdoor environments, where the Kalman filter was used to track the multiple objects and to test the robustness of the proposed system under difficult tracking conditions. Our algorithm demonstrated that contour and global features handling enhances real-time multi-object tracking performance. The results showed that the proposed technique produces better detection rates and data associations. Therefore, our feature work will involve the implementation of the algorithm for tracking multiple fast-moving objects on a huge dataset with more objects such as vehicles.

## Figures and Tables

**Figure 1 sensors-22-02123-f001:**
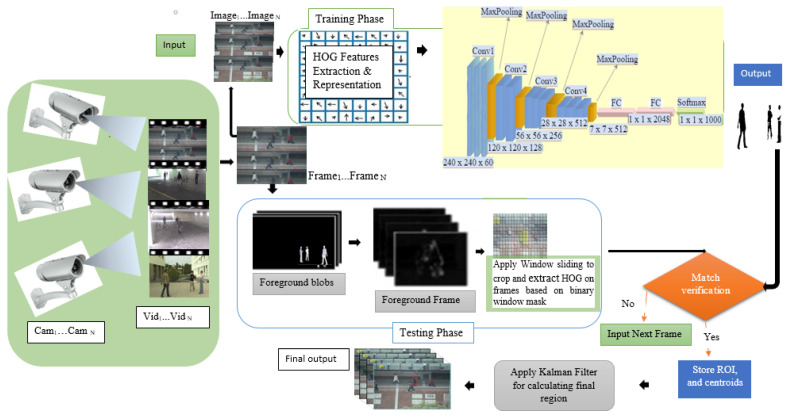
Proposed HCNN method implementation architecture overview.

**Figure 2 sensors-22-02123-f002:**
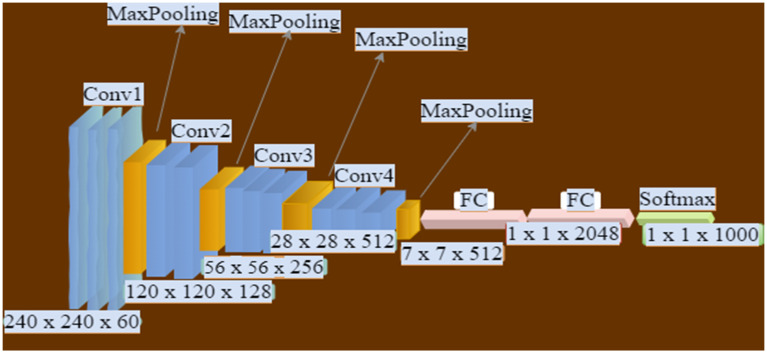
Overview of the CNN structure incorporated into HOG.

**Figure 3 sensors-22-02123-f003:**
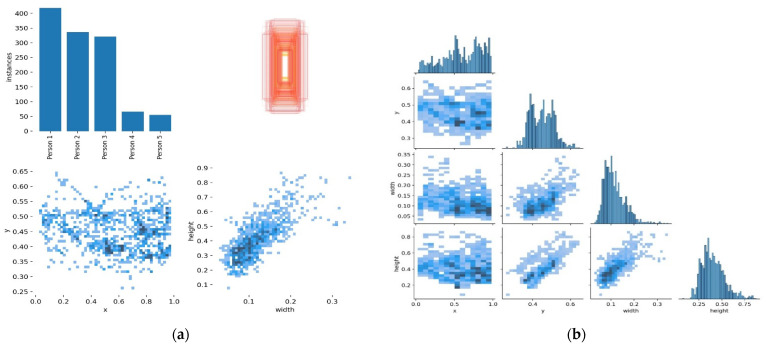
The left (**a**) shows the scatter plot of each instance and label associates, and (**b**) illustrates the correlation relations on the campus scenes images dataset. At most, 90% of the instances are correctly associated with the labels throughout the scenes.

**Figure 4 sensors-22-02123-f004:**
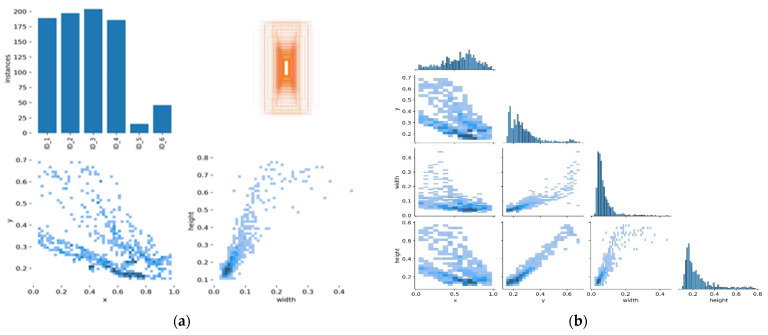
The left (**a**) shows the scatter plot of each instance and label associates, and (**b**) illustrates the correlation relations on the passageway scenes images dataset. Mostly, 92% of the instances are correctly associated with the labels throughout the scenes.

**Figure 5 sensors-22-02123-f005:**
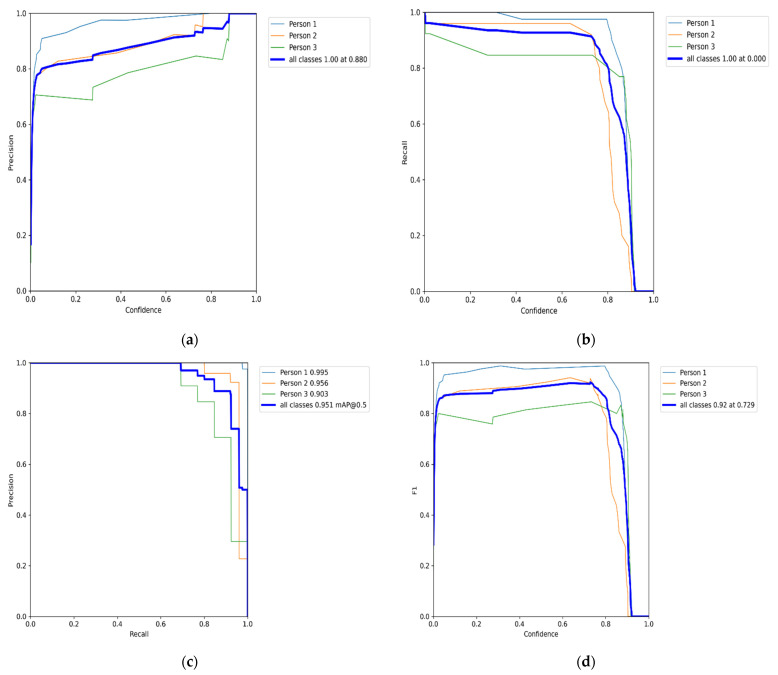
The label (**a**) shows the precision (P) versus confidence (C) graph, (**b**) the recall (R) versus confidence (C), (**c**) is the mean average precision based on comparing the truth bounding box and detection box, and (**d**) the IDF1 score at 92% with confidence of 0.729, advocates the balancing between the P and R based on Campus scenes images dataset. The mAP for all classes is high and accurately modeling detections at 95.1% with a threshold of 0.5. The P and R are high at 88.0%, and 87.5%, respectively, and more confidence at 0.8 and 0.78, respectively, for all classes.

**Figure 6 sensors-22-02123-f006:**
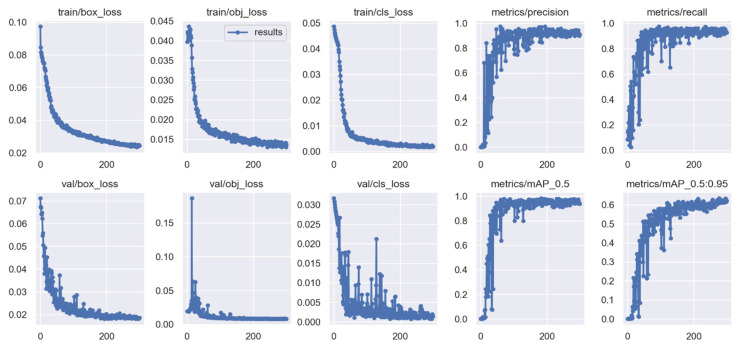
Shows both training and validations losses of the HCNN algorithm’s object detector and classification on 200 epochs for campus scenes dataset. The precision and recall metrics in the training and validation phase converge at the highest of 95.7% accuracy, whereas the mAP converges at 95% with a 0.5 threshold.

**Figure 7 sensors-22-02123-f007:**
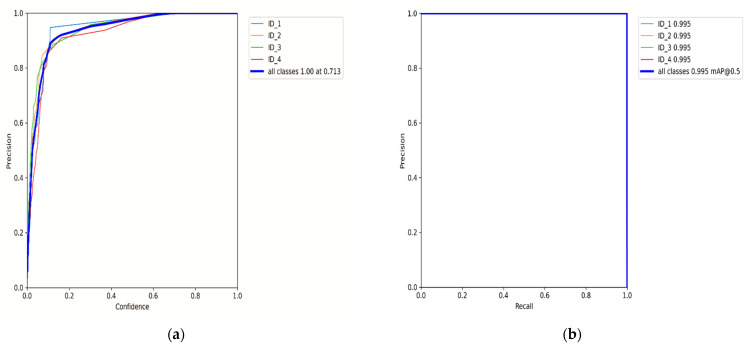
The label (**a**) shows the precision(P) versus confidence(C) graph, (**b**) the recall(R) versus confidence(C), (**c**) is the mean average precision(mAP) based on comparing the truth bounding box and detection box, and (**d**) the IDF1 score at 100% with confidence of 0.626, which advocates the balance between P and R based on passageway scenes dataset. The mAP for all classes is high and accurately modeling detections at 95.1% with a threshold of 0.5. The P and R are high at 100% and 100%, respectively, and more confidence at 0.713 and 0.0 respectively for all classes.

**Figure 8 sensors-22-02123-f008:**
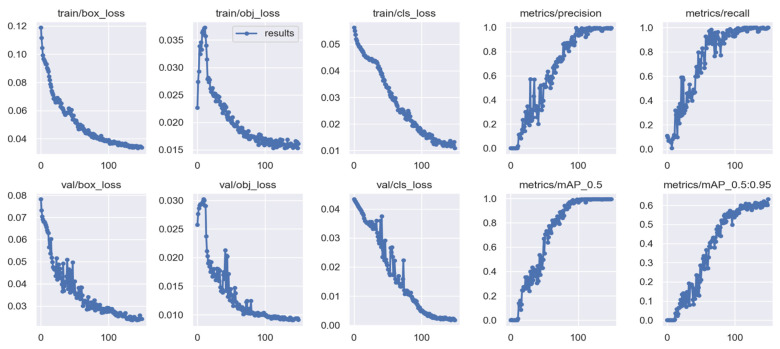
Shows both training and validations of the HCNN algorithm’s object detector and classification loss converging on 100 epochs for passageway scenes dataset. The precision and recall metrics in the training and validation phase converge at the highest of 95.7% accuracy, whereas the mAP converges at 95% with a 0.5 threshold.

**Figure 9 sensors-22-02123-f009:**
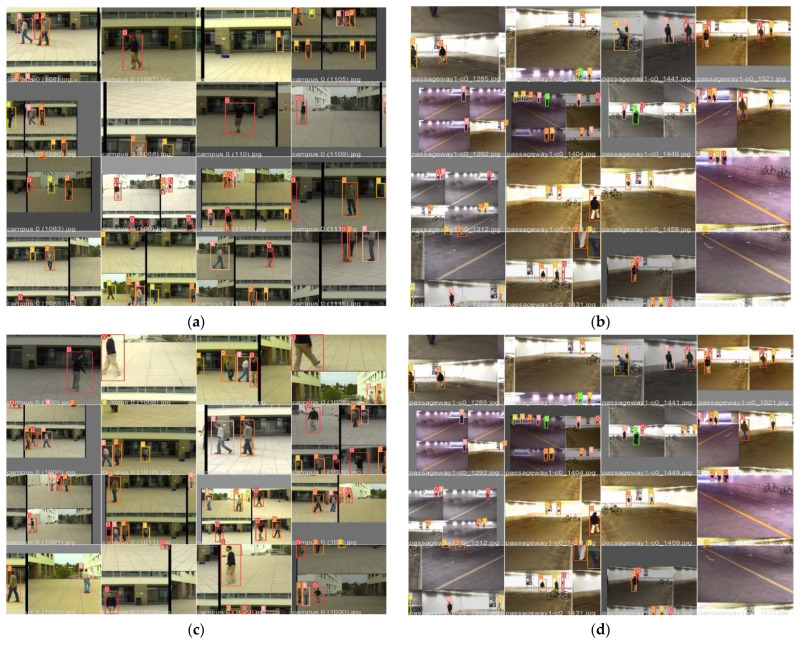
The first row shows visualize (**a**,**b**), the tracking results on validations of both sequences (Campus and Passageway, respectively) with proposed the HCNN algorithm’s tracker. While (**c**,**d**) shows the tracking results of the fine-tuned Yolov5 + Deepsort, (Yolo5Deep) model integrated with HOG and Kalman Filter. (**e**,**f**) shows the EPFL-RCL Multi-cameras frame results for the proposed HCNN without a Kalman Filter and segmentation technique. Compared to our detector and tracker with Yolo5Deep, our proposed algorithm increased positive detections and improved the precision of detection boxes. Moreover, the method is robust for occlusion, illumination, and re-appearance variations.

**Table 1 sensors-22-02123-t001:** Comparison with state-of-the-art methods based on MOT Classification Accuracy.

Methods	Precision ↑	Causality
Improved HOG [4]	86.70%	Online
HOG + 1DCNN [16]	90.23%	Offline
HOG + DCNN Net [32]	96.74	Offline
HOG + CNN [33]	94.14%	Offline
Ours	91.00%	Online

**Table 2 sensors-22-02123-t002:** Performance evaluation metrics on EPFL dataset campus sequence.

Sequences	Precision ↑	Recall ↑	IDF Score ↑	MOTA ↑	MOTP ↑	IDS ↓	ML ↓	MT ↑	FM ↓
CAM#4_scene0	99.4%	96.0%	95.9%	94.0%	91.9%	1	1%	96.0%	2
CAM#4_scene1	98.0%	97.0%	98.0%	93.0%	92.0%	1	1%	94.0%	1
CAM#4_scene2	98.0%	94.0%	96.0%	93.0%	89.0%	2	2%	92.0%	3
CAM#7_scene0	68.0%	80.2%	76.4%	63.3%	75.0%	4	3%	82.0%	5
CAM#7_scene1	88.9%	87.6%	88.2%	83.9%	82.5%	3	2%	88%	2
CAM#7_scene2	95.0%	96.8%	96.3%	90.0%	91.8%	1	1%	92%	1
Overall performance	91.22%	91.93%	91.80%	86.20%	87.03%	2	1.67%	90.67%	3

**Table 3 sensors-22-02123-t003:** Performance evaluation metrics on EPFL dataset passageway sequence.

Sequences	Precision ↑	Recall ↑	IDF Score ↑	MOTA ↑	MOTP ↑	IDS ↓	ML ↓	MT ↑	FM ↓
CAM#1_scene0	94.0%	92.0%	93.0%	89.4%	87.0%	2	2.0%	88.0%	3
CAM#2_scene1	83.0%	82.0%	82.0%	78.0%	76.8%	4	3.0%	86.0%	5
CAM#3_scene2	97.0%	90.8%	93.8%	92.3%	85.8%	2	1.0%	93.0%	2
CAM#4_scene3	76.0%	71.2%	73.5%	71.0%	66.3%	4	4.0%	81.0%	8
Overall performance	87.50%	84.00%	85.58%	82.68%	78.98%	3	2.50%	87.00%	5

**Table 4 sensors-22-02123-t004:** Performance evaluation analysis of fine-tuned Yolo5Deep on EPFL dataset (campus and passageway).

Sequences	Precision ↑	Recall ↑	IDF Score ↑	MOTA ↑	MOTP ↑	IDS ↓	ML ↓	MT ↑	FM ↓
Campus scenes	96.0%	90.6%	91.5%	92.0%	85.0%	2	1.0%	93.0%	2
Passageway scenes	94.0%	92.0%	93.0%	89.4%	87.0%	2	2.0%	88.0%	3

**Table 5 sensors-22-02123-t005:** Performance evaluation analysis of the proposed algorithm without Kalman filter on EPFL-RCL overlapping multi-cameras.

Sequences	Precision ↑	Recall ↑	IDF Score ↑	MOTA ↑	MOTP ↑	IDS ↓	ML ↓	MT ↑	FM ↓
Overall performance	65.0%	56.2%	58.5%	52.0%	46.3%	24	34.0%	54.0%	14

**Table 6 sensors-22-02123-t006:** Comparison with state-of-the-art methods on testing the subset of EPF multi-cameras pedestrian dataset.

Method	MOTA ↑	MOTP ↑	Causality
NCA-Net [32]	64.5%	78.2%	Offline
CNN + HOG Template Matching [11]	94.0%	80.9%	Offline
Yolo + Deepsort [33]	86.1%	88.6%	Online
MCMOT HDM [34]	62.4%	78.2%	Offline
Ours	68.2%	65.0%	Online

## Data Availability

The dataset that resulted from this study can be found at the link: https://www.epfl.ch/labs/cvlab/data/data-pom-index-php/ (last access date: 26 December 2021).

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
