# Peer review of "Enhancing Detection Quality Rate with a Combined HOG and CNN for Real-Time Multiple Object Tracking across Non-Overlapping Multiple Cameras"

_sensors, 2022, doi:10.3390/s22062123_

Round 1
Reviewer 1 Report
The topic is interesting, and useful. The paper is well written, but has some minor drawbacks. Eq. 1 - formatting, please verify eq. 8, as in current form it is not an equation. Please avoid colloquial language, like "we plugged these equations into the Kalman filter". In line 394 you state, that Fig. 3 shows cropped and labeled sample images, but it shows instance and label associations and correlation relations on scenes. In line 461 you state, that the objects are observed and tracked by use of Kalman Filter as shown in Figure 1, while Fig.1 doesn't really show how the Kalman filter is applied. In most part of the text, the Kalman filter is also treated as a "black box".
What I am a little disappointed with is a lack of comparison with other MOT algorithms, and other MOT benchmark datasets (like MOT Challenge, or EPFL-RLC Multi-Camera Dataset).
Also, it is a good practice to always show some statistical importance analysis, when comparing algorithms.
Author Response
Reviewer 1: comments
(a) The topic is interesting, and useful. The paper is well written, but has some minor drawbacks. Eq. 1 - formatting, please verify eq. 8, as in current form it is not an equation. Please avoid colloquial language, like "we plugged these equations into the Kalman filter". In line 394 you state, that Fig. 3 shows cropped and labeled sample images, but it shows instance and label associations and correlation relations on scenes. In line 461 you state, that the objects are observed and tracked by use of Kalman Filter as shown in Figure 1, while Fig.1 doesn't really show how the Kalman filter is applied. In most part of the text, the Kalman filter is also treated as a "black box".
Authors’ Response: We proofread the whole paper, re-constructed, and re-clarified the sentences to relay the effective information. We further book and send the paper to the professional English proofreading and Editing Company (Busy Bee Pty Ltd). Therefore, the typo errors, misspellings, and unclear sentences that hindered the relay of the intended information were addressed in those two phases (Our proofreading and Editing, and the professional English proofreading and editing service (The proofreading and editing certificate issued)). The wrong reference sentencing in line 394 is corrected. Fig 1. has always shown the implementation of the Kalman filter, however, to make it more readable, visible, and clear we redrew fig 1. The equation format 8 is corrected.
Reviewer 1: comments
(b) What I am a little disappointed with is a lack of comparison with other MOT algorithms, and other MOT benchmark datasets (like MOT Challenge, or EPFL-RLC Multi-Camera Dataset). Also, it is a good practice to always show some statistical importance analysis, when comparing algorithms.
Authors’ Response: We did the comparison of the studies (with the fine-tuned Yolov5+Deepsort and integrated KF). We used the same dataset with the YoloV5+deepsort tracker model and integrated the Kalman Filter to extract the spatial information from the bounded box object and pass them to the Hog feature extractor, which sent them2D array to the CNN for flattening and further modification of features. The evaluation results/statistical analysis are displayed in Table 4. The visual results comparison is discussed and displayed in fig 9.
- We further visualized and compared the qualitative results in Figure 9. We tested our algorithm on the EPFL pedestrians' dataset and showed qualitative results comparisons. The ablation studies on the proposed algorithm are done on the EPFL-RLC Multi-Camera dataset set and the qualitative results are demonstrated in figure 9, whereas the statistical results are displayed in table 5. We’ve added Table 6 and compared our algorithms to several state-of-the-art approaches, both online and offline versions.

Reviewer 2 Report
This paper studies the object tracking problem using the HOG features for CNN. All applied techniques are classical algorithms from the literature. Thus I think this paper has limited novelty. Moreover, the proposed algorithm is not very clearly presented.
- Many grammar errors and typos. For example, in the caption of figure 9, the performances are shown in Tables 2 and 3, not 1 and 2 as stated. I would recommend the authors proofread the whole manuscript before submission.
- The proposed algorithm is not described very clearly in Section 3. For example, what are the meaning of $N$ and $K$ in figure 1? Figure 1 shows that the output of the CNN is a color image. In line 22, the system input is an image of size 80x32 but it becomes 120x32 in line 171. Does CNN process images from different cameras at the same time or one image from one camera at a time or just one object of interest? What is the format of the CNN output? How the match verification is done?
- I think the ablation experiment would be helpful to demonstrate the effectiveness of each component in the system. I suspect the background subtraction plays the most vital role in this project.
Author Response
Reviewer 2: Comments
(a). This paper studies the object tracking problem using the HOG features for CNN. All applied techniques are classical algorithms from the literature. Thus, I think this paper has limited novelty. Moreover, the proposed algorithm is not very clearly presented.
Authors’ Response: We proofread the whole paper, re-constructed, and re-clarified the sentences to relay the effective information. We further book and send the paper to the professional English proofreading and Editing Company (Busy Bee Pty Ltd). Therefore, the typo errors, misspellings, and unclear sentences (illogical sentences and phrases) that hindered the relay of the intended information were addressed in those two phases (Our proofreading and Editing, and the professional English proofreading and editing service (The proofreading and editing certificate issued). Moreover, in Section 3, the proposed algorithm implementation is detailed robustly into five subtopics.
Reviewer 2: Comments
- Many grammar errors and typos. For example, in the caption of figure 9, the performances are shown in Tables 2 and 3, not 1 and 2 as stated. I would recommend the authors proofread the whole manuscript before submission.
Authors’ Response: We proofread the whole paper, re-constructed, and re-clarified the sentences to relay the effective information. We further book and send the paper for the professional English proofreading and Editing Company (Busy Bee Pty Ltd). Therefore, the typo errors, misspellings, and unclear sentences (illogical sentences and phrases) that hindered the relay of the intended information were addressed in those two phases (Our proofreading and Editing, and the professional English proofreading and editing service (The proofreading and editing certificate issued)). Hence, the caption of Fig 9 is corrected.
Reviewer 2: Comments
- The proposed algorithm is not described very clearly in Section 3. For example, what are the meaning of $N$ and $K$ in figure 1? Figure 1 shows that the output of the CNN is a color image. In line 22, the system input is an image of size 80x32 but it becomes 120x32 in line 171. Does CNN process images from different cameras at the same time or one image from one camera at a time or just one object of interest? What is the format of the CNN output? How the match verification is done?
Authors' Response: The typo errors and unclear sentences (illogical sentences and phrases) that hindered the relay of the intended information were addressed in those two phases (Our proofreading and Editing, and the professional English proofreading and editing service (The proofreading and editing certificate issued). Moreover, in Section 3, the proposed algorithm implementation is detailed robustly into five subtopics. The image size typo error($N$ and $K$) was also corrected. The input intake is well defined in section 3, and Algorithm takes multiple images from multiple cameras. It processes these images as a batch and feature vectors are fed into the CNN for further process and verification. Therefore, the output of CNN is the characteristics information of the target(object). The matching details are detailed well in subsections (3.1-3.2).
Reviewer 2: Comments
- I think the ablation experiment would be helpful to demonstrate the effectiveness of each component in the system. I suspect the background subtraction plays the most vital role in this project.
Authors' Response: We did the comparison of the ablation studies using the same dataset with the YoloV5+deepsort tracker model and integrated the Kalman Filter to extract the spatial information from the bounded box object and pass them to the Hog feature extractor which sent them2D array to the CNN for flattening and further modification of features. The evaluation results/statistical analysis are displayed in Table 4. The visual results comparison is discussed and displayed in fig 9.
- We further visualized and compared the qualitative results in Figure 9. We tested our algorithm on the EPFL the EPFL-pedestrians' dataset and showed the tracking results of the comparisons of the qualitative results find-tuned Yolov5 + Deepsort (Yolo5Deep) model integrated with HOG and Kalman Filter. The ablation studies on the proposed algorithm are done the EPFL-RCL Multi-Camera on the proposed HCNN algorithm without a Kalman Filter EPFL-RLC Multi-Camera dataset set and the qualitative results are demonstrated in figure 9, whereas the statistical results are displayed in table 5. We’ve added Table 6 and compared our algorithms to several state-of-the-art approaches, both online and offline versions

Reviewer 3 Report
. Image quality of Fig. 1 has low. The author should be needs to rework the Fig. 1 for the improvement.
. The contribution has soundness, but the persuasive strength has weak. It should be improved by comparing the author's proposal to the opposite side scenario in the figure.
. It appears that Fig. 9 should be compared to other algorithms. It's difficult to know if the outcomes are truly excellent as the author`s proposed contribution.
. It looks like the numbers in Table. 9 are obscured. This part needs to be corrected.
. Fig. 5 and Appendix do not match. This part needs to be corrected.
. The description of the Kalman filter is very general, and unless there is a specific contribution, this section should be summarized or replaced with the reference.
Author Response
Reviewer 3: Comments
- Image quality of Fig. 1 has low. The author should be needs to rework the Fig. 1 for the improvement.
Authors’ Response: We proofread the whole paper, re-constructed, and re-clarified the sentences to relay the effective information. We further book and send the paper to the professional English proofreading and Editing Company (Busy Bee Pty Ltd). To make Figure1 more visible, readable, and clear we redrew it.
Reviewer 3: Comments
b. The contribution has soundness, but the persuasive strength has weak. It should be improved by comparing the author's proposal to the opposite side scenario in the figure.
c. It appears that Fig. 9 should be compared to other algorithms. It's difficult to know if the outcomes are truly excellent as the author`s proposed contribution.
Authors’ Response: for comments (b) and (c); we proofread the whole paper, re-constructed, and re-clarified the sentences to relay the effective information. We further book and send the paper to the professional English proofreading and Editing Company (Busy Bee Pty Ltd). Therefore, the typo errors, misspellings, and unclear sentences that hindered the relay of the intended information were addressed in those two phases (Our proofreading and Editing, and the professional English proofreading and editing service (The proofreading and editing certificate issued)). We further did the comparison of the ablation studies using the same dataset with the YoloV5+deepsort tracker model and integrated the Kalman Filter to extract the spatial information from the bounded box object and pass them to the Hog feature extractor which sent them2D array to the CNN for flattening and further modification of features. The evaluation results/statistical analysis are tabled in Table 4. The visual results comparison is discussed and displayed in fig 9.
- We further visualized and compared the qualitative results in Figure 9. We tested our algorithm on the EPFL the EPFL-pedestrians' dataset and showed the tracking results of the qualitative results comparisons fine-tuned Yolov5 + Deepsort (Yolo5Deep) model integrated with HOG and Kalman Filter. The ablation studies on the proposed algorithm are done the EPFL-RCL Multi-Camera on the proposed HCNN algorithm without a Kalman Filter EPFL-RLC Multi-Camera dataset set and the qualitative results are demonstrated in figure 9, whereas the statistical results are displayed in table 5. We’ve added Table 6 and compared our algorithms to several state-of-the-art approaches, both online and offline versions
Reviewer 3: Comments
d. It looks like the numbers in Table. 9 are obscured. This part needs to be corrected.
Authors’ Response: We searched in our script for table 9, but could not understand what the reviewer is referring to here since our manuscript has only Tables 1-3. However, all the contents or numbers in our tables are well adjusted as per the journal author’s guidelines
Reviewer 3: Comments
e. Fig. 5 and Appendix do not match. This part needs to be corrected
Authors’ Response: We could not find the mismatch. However to ensure that we address the reviewer’s concern we re-plotted the confusion matrix and the statistical results (P, IDF1, and R) graphs.
Reviewer 3: Comments
f. The description of the Kalman filter is very general, and unless there is a specific contribution, this section should be summarized or replaced with the reference.
Authors’ Response: We increase the citation in the Kalman filter section and ensured more referencing of the pieces of literature that defined and used the Kalman filter for object tracking.

Round 2
Reviewer 2 Report
I think the revised version is much better than the original version. Some minor comments.
- What is the meaning of 'Training Phase' and 'Testing Phase' in Figure 1?
- The subplots are not cited correctly in the caption of Figure 9. '(c) and (b)' or '(c) and (d)'?
- I appreciate the newly added Section 5.1 and Table 4 provides a very good comparison. But the ablation study corresponds to comparing the proposed method with itself with missing components. For example, YOLO+KarmanFilter vs ProposedMethod+NoKarmanFilter is not an ablation study. ProposedMethod+NoKarmanFilter vs ProposedMethod +KarmanFilter is an ablation study for the Karman filter component in your method.
Author Response
Round 2: Reviewer 2’s comments:
- Spellings and typos are checked and fine.
Authors' response: We proofread the whole manuscript and rigorously checked and corrected all the typo errors and spellings. We further sent the manuscript for the English editing services.
- What is the meaning of 'Training Phase' and 'Testing Phase' in Figure 1?
Authors response: Training Phase is where the basic training of our proposed algorithm. This is where we enable the algorithm to modify the training parameters and use the input data to create output. Testing Phase is where we fed our algorithm with the live data. It is where we are evaluating the input data live and putting out live results
- The subplots are not cited correctly in the caption of Figure 9. '(c) and (b)' or '(c) and (d)'?
Authors Response: The subplots citation in the caption of figure 9 is corrected, (c) and (b) were replaced with (c) and (d).
- I appreciate the newly added Section 5.1 and Table 4 provides a very good comparison. But the ablation study corresponds to comparing the proposed method with itself with missing components. For example, YOLO+KarmanFilter vs ProposedMethod+NoKarmanFilter is not an ablation study. ProposedMethod+NoKarmanFilter vs ProposedMethod +KarmanFilter is an ablation study for the Karman filter component in your method.
Authors Response: The Ablation study is conducted with the EPFL multi-camera dataset on our algorithm with no Kalman filter. The results were recorded and displayed in figure 9 (e) and (f) and tabled in Table 5. They were further compared, discussed, and analysed in Section 5 (Results Analysis).
